# Manufacture and Property of Warp-Knitted Fabrics with Polylactic Acid Multifilament

**DOI:** 10.3390/polym11010065

**Published:** 2019-01-04

**Authors:** Tong Yang, Wei Zhou, Pibo Ma

**Affiliations:** 1Engineering Research Center for Knitting Technology, Ministry of Education, Jiangnan University, Wuxi 214122, China; yangtong0305@163.com; 2College of Textile and Clothing, Jiangnan University, Wuxi 214122, China; zhouwei7690@163.com; 3State Key Laboratory of Bio-Fibers and Eco-Textiles, Qingdao University, Qingdao 266071, China

**Keywords:** polylactic acid multifilament, warp-knitted fabric, dye property, mechanical property, thermal and moisture comfort property

## Abstract

This study investigates the properties of polylactic acid (PLA) multifilament and its warp-knitted fabrics. Multifilament properties were tested and compared with PET multifilament with different diameters. The 83.3 dtex PLA multifilament was used to knit the fabric, and the fabric properties before and after dyeing were studied. Results showed that the mechanical properties of PLA multifilament were comparable to those of PET. However, PLA had a higher heat shrinkage rate. The dyed PLA warp-knitted fabric has excellent color fastness. Due to the influence of temperature and dye particles during the dyeing process, the breaking strength, air permeability and moisture permeability of the fabric were decreased. On the contrary, the elongation at break, abrasion resistance, anti-pilling properties, drape and crochet value of the fabric were increased.

## 1. Introduction

As a kind of bio-based synthetic fiber, polylactic acid (PLA) fiber is an environment-friendly synthetic polymer material. It is produced by a melt spinning technique after fermentation and polymerization of raw materials such as corn and cereals [1]. Energy depletion and the non-degradability of waste from petroleum products pose threats to the environment with the rapid growth of petroleum-based synthetic fiber production. PLA draws more considerable interest due to its degradability, as well as excellent mechanical properties, appearance gloss, moisture permeability, flame retardancy, UV stability, antifouling and antibacterial property. PLA fiber can be made into monofilament, multifilament, short fiber, false twist textured yarn, woven fabric and non-woven fabric, which make it widely applied in clothing and industrial fields, such as weft knitted underwear fabrics, sanitary textiles, medical textiles and agricultural textiles, etc. Analysis and summary of the properties of multifilament and warp-knitted fabrics can provide a theoretical reference for the subsequent development of PLA.

Lim L-T et al. studied the surface and mechanical properties of weft-knitted fabrics made from PLA staple fibers. It was found that the values for bending stiffness, bending lag, shear stiffness and shear lag of the PLA staple fiber fabric were significantly decreased after the washing treatment [1,2]. Anakabe et al. studied the PLA/PMMA blend compared to pure PLA and found that the mechanical properties were improved [3]. Gerard and Zembouai et al. studied the morphology and mechanical properties of PLA/PHBV blends, and Huang et al. studied the properties of weft knits of PLA/PHBV blended yarns. The results showed that the PLA/PHBV yarn woven fabric had excellent antibacterial properties [4,5,6,7]. Zhang et al. studied the dyeing process of PLA and PHBV blends. The experimental results showed that an excellent dyeing effect and bursting strength can be achieved by properly applied dyes (e.g., C.I. Disperse Orange 30, Red 74, and Blue 79) under optimal low-dyeing-temperature conditions (100 °C, 10 min, pH 5, LR 30: I) [8,9,10]. Ramesh and Hakim et al. also studied the properties of PLA blends, which are electrospinning scaffolds of PLA/CS/TPC and composites of PLA/SiO_2_ [11,12]. Avinc et al. studied the effects of flame retardants and oil/water repellent finishes on the properties and color of PLA staple fabrics. It was found that the softener/lubricant used in the process has a detrimental effect on the oil recovery performance of the fabric after hot pressing [13,14,15]. Currently, most of the existing research on PLA focuses on blends of PLA and PHBV, and its woven fabrics. Compared with other structures, warp-knitted fabrics have stronger anti-dissociation ability and more efficient production speed, which is easier to industrialize. However, no research has involved the warp-knitting fabrics of PLA multifilament in application.

The research described in this paper aims to study the properties of PLA multifilament and warp-knitted fabrics. PLA multifilament with different diameters was studied for its properties and warp-knitted fabrics with different dyeing states were compared to study the effect of the fabric finishing process on properties. This provides a new idea for how to promote the application of new textile materials and bring new enlightenment to the development of new materials.

## 2. Materials and Experimental Section

### 2.1. Warp-Knitted Fabric Manufacture

The 83.3 dtex/36 f PLA multifilament was used to knit fabrics of velveteen structure, and the 3D virtual display of this structure is shown in Figure 1. Warp-knitted fabrics were produced on the HKS2 type of warp-knitting machine (German Karl Mayer). The machine gauge was E28, as shown in Figure 2, and the actual fabric’s picture is marked by the red square. Table 1 shows the structural parameters of warp-knitted fabrics.

PLA has a large number of ester and methyl groups in its chemical structure, with no hydrophilic polar or reaction groups. Ordinary dye cannot be normally used because of its tight molecular structure and low macromolecule. For this reason, the disperse dye GS/red was chosen to dye PLA multifilament in this experiment. It has good diffusibility, stable dispersion and linear molecules.

The fabric was dyed and subjected to a post-cleaning treatment in accordance with the dyeing process of Figure 3. In the figure, a represents a dye, b represents a leveling agent, and c represents a fabric. The dyeing was carried out in a pH of 4.5. When the water temperature reached 40 °C, 2% of disperse dye GS/red #153 (Zhejiang Hawthorn Dye Chemical Co., Ltd., Taizhou, China), 0.1/g·L^−1^ HAC and 1% leveling agent were sequentially added, and then finally the fabric was added. The bath ratio was 1:20. The temperature was raised to 110 °C at a rate of 1 °C/min and then maintained for 30 min to allow the fabric and dye solution to react. The temperature was then reduced to 60 °C in a natural state. The reduction cleaning was carried out in an environment of 60–65 °C for 15 min in the solution of 2 g·L^−1^ Na_2_CO_3_ and 2 g·L^−1^ insurance powder.

### 2.2. Mechanical Properties of PLA Multifilament

In order to study the structure and properties of PLA multifilament, 55.6 dtex, 83.3 dtex PLA and 83.3 dtex polyester (PET) multifilament were compared to determine the properties of PLA.

Infrared spectra was conducted on the Nicolet iS10 Fourier Infrared (FT. IR) spectrometer (Thermo Fisher Scientific Co., Ltd., New York, NY, United States) to qualitatively analyze the composition of PLA. The collection range was 4000–500 cm^−1^. 

SEM of two kinds of multifilament was sliced in the longitudinal axis and in cross-section section with a slicer. They were fixed on a metal sample stage with conductive tape for gold spraying. SU510 type scanning electron microscope (Hitachi, Ltd., Hitachi, Japan) was selected to observe the transverse and longitudinal forms of multifilament.

Tensile strength and elongation at break were tested on a YG020B electronic single yarn strength machine (Changzhou Second Textile Machinery Co., Ltd., Changzhou, China), with 500 mm/min test speed and 250 mm fixed length. The results were averaged over 10 times.

Moisture absorption experiment was carried out in an actual moisture regain rate at 20 °C and a relative 54% humidity. The multifilament was dried in a fast eight-barrel oven at 105 °C for 30 min until equilibrium. During the heating process, the mass was weighed every 5 min. The moisture regain of the multifilament was calculated according to the formula as follows:(1)$$W=\frac{G-G_{0}}{G_{0}}\times 100\%$$

In Formula (1): *W* is the moisture regain of the filament; *G* and *G*_0_ respectively represent the wet weight and dry weight of the filament, and the unit is g.

Thermal shrinkage rate of PLA multifilament was investigated at different temperatures and compared with PET multifilament. Ten sets of samples for each multifilament were respectively placed in a beaker containing deionized water. They were treated for 25 min at 40, 60, 80 and 100 °C water bath temperatures.

### 2.3. Mechanical Properties of Warp-Knitted Fabric

Tensile strength of the fabrics was tested on the YG026A electronic fabric strength meter (Changzhou Second Textile Machinery Co., Ltd., Changzhou, China). Before the experiment, the elongation at break of the fabric was tested to determine the approximate elongation range, the appropriate holding and the tensile speed for test. The holding distance and the stretching speed were respectively set to 200 mm and 100 mm/min.

Abrasion resistance was conducted on the Y522 Taber Fabric Wear Tester (Changzhou Second Textile Machinery Co., Ltd., Changzhou, China). The pressurized weight was set to 250 g. Each sample was ground 160 times to compare the degree of wear.

Anti-pilling property was measured on the YG502N fabric pilling machine (Changzhou Second Textile Machinery Co., Ltd., Changzhou, China). The weight of Heavy hammer was set to 290 cN. The nylon brush and abrasive fabric were individually rubbed 50 times. Results came from a comparison with the five-standard sample.

Drape was tested on the YG811E photoelectric fabric drape tester (Changzhou Second Textile Machinery Co., Ltd., Changzhou, China). We chose a rotation speed of 30 r/min for the experiment, testing the drape properties of the fabric in both dynamic and static states.

Breathability was tested on the YG(B)461E-III fully automatic gas permeability meter (Ningbo Textile Instrument Factory, Ningbo, China). The testing pressure difference was 100 Pa, the testing area was 20 cm^2^. Each sample was tested 10 times for averaging.

Warmth property was measured on the YG606D flat fabric insulation meter (Nantong Sansi Electromechanical Technology Co., Ltd., Nantong, China).

Moisture permeability test was conducted on the YG601H-II computerized fabric moisture permeable instrument (Shanghai Xusai Instrument Co., Ltd., Shanghai, China) at 90% humidity. The CaCl_2_ was combined in a moisture permeable cup containing anhydrous fabric in a moisture permeable device of constant 38 °C, which was dried at 160 °C for 3 h in the HD101A electric blast oven. The sample was weighed after drying for 30 minutes in the presence of dry silica. 

### 2.4. Dyeing Properties of Warp-Knitted Fabrics

Percentage of dyeing was measured by a 721 spectrophotometer. The formula for calculating the percentage of dyeing *A*% is as follows:(2)$$A\%=\frac{A_{0}\times 30-A_{1}\times 8}{A_{0}\times 30}$$

In Formula (2): *A*_0_ is the absorbance of the dye solution before dyeing, and *A*_1_ is the absorbance of the dye residue after dyeing. 

A magnetoresistance test for warp-knitted fabrics was conducted on a Datacolorspectraash spectrophotometer (Datacolor 650, Lawrenceville, NJ, USA) with a 10° standard viewing mirror. The Kubelka-Munk equation is used to calculate the color intensity based on the magnetoresistance value, which is as follows:(3)$$K/S=\frac{1-R^{2}}{2R}$$

In Formula (3): K is the absorption coefficient, *S* is the scattering coefficient, and *R* is the magnetic reluctance of the sample at a given wavelength: 512 nm.

The washing color fastness of warp-knitted fabrics was tested on a SW-24E wash fastness tester (Wenzhou Darong Textile Instrument Co.,Ltd., Wenzhou, China). Before testing, the sample cloth was sewn together with cotton and wool clothes. The sample was washed in the solution with 5 g/L of standard soapy water and 2 g/L of Na_2_CO_3_ at a temperature of 50 °C for 45 min. The sample cloth was dried at 60 °C, and the degree of dyeing was evaluated against a gray sample card.

Sublimation color fastness was conducted on a YG (B) 605 type ironing sublimation color fastness tester. The PLA fabric and cotton fabric were cut into rectangles of 4 cm × 10 cm in accordance with the specifications of the samples. They were then pressed onto the instrument at a temperature of 80 °C for 20 s. The degree of dyeing was evaluated and compared against a gray sample card.

## 3. Results and Discussions

### 3.1. Properties of PLA Multifilament

The infrared spectrum of PLA is shown in Figure 4. According to the position and intensity of the absorption wavelength, it was found that the strongest peak in the spectrum was 1750 cm^−1^, which is the C=O stretching vibration peak. The stretching and bending vibration peaks of —CH and —CH_3_ at 2923.51 and 1454.97 cm^−1^, and the stretching vibration peak of C—O—C at 1177.88, 1129.98 and 1083.69 cm^−1^, indicate there is the presence of an ester group.

The methyl group present on the molecular chain of PLA leads it to crystallize easily (the molecular chain is strong and the side chain is short). However, the melting point and *T*_g_ are low and sensitive to temperature because of the lower intermolecular force.

SEM images of longitudinal section and cross-section morphology of two kinds of multifilament are shown in Figure 5. In the longitudinal direction, both kinds of multifilament are displayed in a neatly arranged column, which have a high orientation degree and crystallinity. That is the reason for poor hygroscopicity and dyeing difficulties of the PLA multifilament. While in comparison, the surface of the PLA has more impurities, which is mainly due to the influence of the spinning process. On the cross-section, two types of multifilament with a compact structure are nearly circular in shape.

The mechanical properties of three kinds of multifilament are shown in Table 2. As can be seen, the tensile strength of PLA is close to that of PET with the same diameters, which has approximative crystallinity, contributing to its resistance to stretching. In the process of tensile fracture, the macromolecular chain of the folded portion is first straightened, and then the mutual linkage between the molecule and the other molecule is started. Due to the high crystallinity of PLA, the molecular structure is not easily destroyed. That is the reason why PLA multifilament has better mechanical properties. Relatedly, the elongation at break of PET multifilament is greater, that is, the elasticity of PET is better. It is mainly due to the differences in macromolecular structure and the bond length of the chemical bond between two multifilaments. In addition, when the diameter of the PLA multifilament decreases, both the breaking strength and the breaking strength decrease.

The hygroscopic property test results of PLA and PET multifilament at room temperature are shown in Table 3. According to the data, it was found that PLA multifilament has a higher moisture regain comparing with PET multifilament. However, the hygroscopic property of PLA multifilament is still poor. This is mainly caused by the macromolecular structure of PLA, which does not contain other hydrophilic groups in the molecule except for one hydroxyl group at each end of the macromolecule chain. The arrangement of the PLA macromolecular chains is relatively regular, leading to high crystallinity, the pores between the molecular chains are smaller, which makes it resistant to moisture ingress.

Thermal shrinkage rates of the multifilament are shown in Table 4, and a processed line chart is shown in Figure 6. It can be seen that the 83.3 dtex PET multifilament had almost no shrinkage after being treated in a water bath at different temperatures. The shrinkage rate of PLA multifilament increased with the increasing of temperature, and the shrinkage rates of PLA with two kinds of diameters were 30–40 times that of PET. This is mainly due to the fact that PLA has a lower glass transition temperature, resulting in contraction when heated. Other than this, PLA multifilament of different thicknesses differed in temperature, at which sharp shrinkage began under heated conditions. When the temperature at which it began to shrink sharply raised, the diameter of the multifilament became thicker, which was mainly because the heat deformation was more pronounced in the finer fibers and only appeared in the thicker fibers over a period of time.

With the same conditions, when the temperature does not exceed 80 °C, the shrinkage rate of 83.3 dtex PLA multifilament was lower compared with 55.6 dtex PLA multifilament. Conversely, the rising trend of the shrinkage rate of both multifilament tended to be same. However, as the diameter of the multifilament decreased, the heat shrinkage rate increased by about 2%. This conclusion aims to better formulate the specifications of the fabric in practical applications.

### 3.2. Properties of Warp-Knitted Fabrics

The mechanical properties of warp-knitted fabrics before and after dyeing were evaluated by breaking strength and elongation, as shown in Table 5. It can be seen from the data that the breaking strength after dyeing reduced about 16%, while the elongation at break increased by about 67%. This is consistent with the previous test of the multifilament. The main reason for the decrease in the breaking strength was that the temperature was up to 110 °C during dyeing, which caused shrinkage of the multifilament.

With the multifilament being subjected to heating, macromolecular chains exacerbate movement, especially in macromolecules where amorphous regions begin to produce dislocation slip. It leads to a decrease in the alignment of the macromolecules in the multifilament, and it is more likely to break when stressed. The degree of orientation of the filament is inversely proportional to the elongation at break. Therefore, the degree of orientation of the multifilament increases with the elongation decreasing.

The abrasion resistance of warp-knitted fabrics was measured by rubbing the sample according to the geometrical trajectory. Worn samples are shown in Figure 7. It was found that the undyed PLA fabric showed multiple holes after the friction test. However, the dyed fabric has only the same friction marks as the shape of the friction track, and no hole was found.

Silicone softener is generally added to make fabrics feel smoother and softer during the dyeing. Therefore, it was supposed that the abrasion resistance of the fabric would be improved because of the addition of silicone softener. In order to confirm this conjecture, we tested the abrasion resistance of the dyed fabric with no silicone softener. The degree of wear is shown in Figure 8. As can be seen, the fabric has fewer holes after rubbing compared with the undyed fabric.

There are two main factors that aim to the improvement of the abrasion resistance of dyed fabrics. The first is the addition of silicone softener during the dyeing process. Silicon softeners reduce the surface roughness of the fabric, making it smoother, which relieves the degree of damage during friction. The second is disperse dyes. Disperse dye coats the outside of multifilament in the form of particles. When the fabric is subject to wear, the pressurized weight has to wear through the dye particles to touch the fabric.

The anti-pilling property is graded according to the five-level standard. Fabrics have better property with the increasing rating. The dyed fabric is level 5, and the undyed fabric is level 4.5. Sample pictures after testing are shown in Figure 9. It can be seen that that the undyed fabric showed slight fluff and the pilling phenomenon was not very noticeable. The dyed fabric was almost unaffected by the test. The anti-pilling property of filament fabrics is generally better and goes further with the addition of silicone softener during the dyeing process, which makes the fabric surface smoother and with less pilling.

Drapability is generally expressed by the drape coefficient in static or dynamic. The drape of fabrics becomes better as the drape coefficient decreases. The test results are shown in Table 6, and the sample morphology is shown in Figure 10. It was found that dyed fabrics have a smaller drape coefficient in both static and dynamic, which is mainly due to the addition of silicone softener during the dyeing.

Breathability, warmth and moisture permeability are usually used to evaluate fabric application properties. Results are shown in Table 7.

The Crowe value is related to the comfort that the human body feels in a particular environment. When the Crowe value exceeds 1, the body feels warm. On the contrary, cold feeling will be passed to the body. As can be seen, the fabrics before and after dyeing were similar in value, which means they had a strong sense of coldness. At the same time, the fabric had a lower heat retention coefficient, due to the thermal conductivity of the PLA. 

The moisture permeability of the fabric is generally related to the hygroscopicity of the material and the texture of the fabric. PLA fabrics have a slightly higher moisture permeability compared with conventional chemical multifilament. As shown in Table 7, the moisture permeability of the dyed fabric decreased because of the dye particles forming a coating on the multifilament surface, which reduced the pores of the fabric structure. 

The fabric has better breathability with the increasing of air permeability. It was found that the air permeability of the dyed fabric was reduced by about 33%, which caused a reduction in fabric porosity because of the coating of the disperse dye particles. 

### 3.3. Dyeing Properties of Warp-Knitted Fabrics

Percentage of dyeing was measured through experiments: *A*_0_ is 1.687 and *A*_1_ is 0.87. After calculation, the percentage of dyeing of PLA fabric was *A*% = 48.80%.

Magnetoresistance of the fabrics was calculated: The K/S value before dyeing was 26.888, and the K/S value after dyeing was 21.307. The fixing rate of PLA fabric was 46.4%.

Washing color fastness were as follows:(1)PLA cloth Decoloration fastness: 4.15, Level 4;(2)Cotton cloth color fastness: 4.82, Level 5;(3)Wool cloth Color fastness: 4.10, Level 4.

Sublimation color fastness test results were level 5, this is the highest level. Combining two color fastness test datum, it was found that the warp-knitted fabrics of PLA multifilament dyed with disperse dyes had a higher color fastness. There are many ester groups in the molecular chain of the PLA, which is the reason why PLA fabrics have better color fastness. The ester groups have a higher affinity with the ester groups, hydroxyl groups and halogen atoms in the dye molecules, contributing to their diffusion into the fibers and bonding to the fiber molecules easily through dipole forces or hydrogen bonds.

## 4. Conclusions

This paper reports the manufacturing process and the properties of warp-knitted fabric with PLA multifilament, with respect to its microchemical structure, surface morphology, mechanical properties and heat and humidity properties. The following conclusions are drawn: (1)The mechanical properties of PLA multifilament are comparable to PET, apart from the poor heat shrinkage rate. We aim to adjust process parameters more reasonably in further use.(2)The dyeing process has a great influence on the mechanical properties of the PLA warp-knitted fabrics. During the production, special attention should be paid to the control of the temperature during dyeing and finishing, as well as the damage to the performance of the PLA multifilament. However, soft touch, abrasion resistance, anti-pilling performance and application properties of fabrics have been improved.

## Figures and Tables

**Figure 1 polymers-11-00065-f001:**
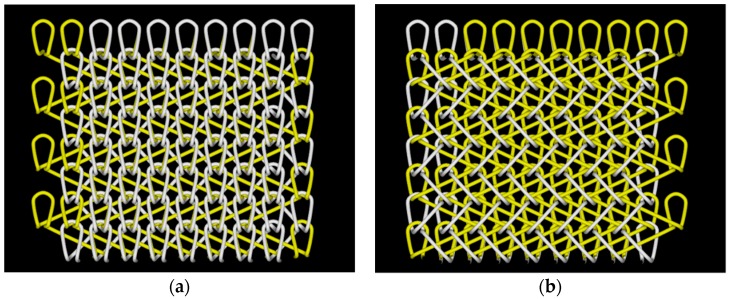
3D virtual display of warp-knitted fabrics: (**a**) fabric front; (**b**) reverse side of fabric.

**Figure 2 polymers-11-00065-f002:**
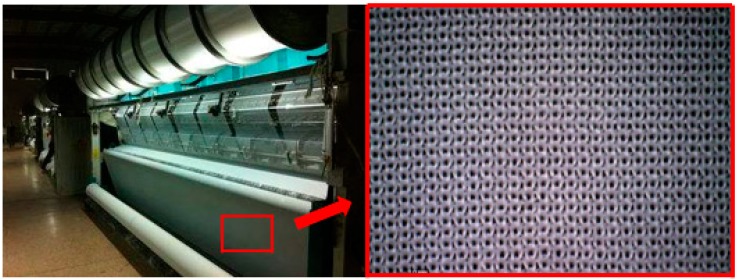
Fabrics knitted on the warp-knitted machine.

**Figure 3 polymers-11-00065-f003:**
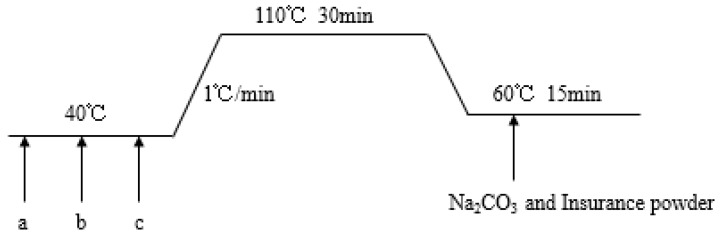
Dyeing flow chart of warp-knitted fabrics.

**Figure 4 polymers-11-00065-f004:**
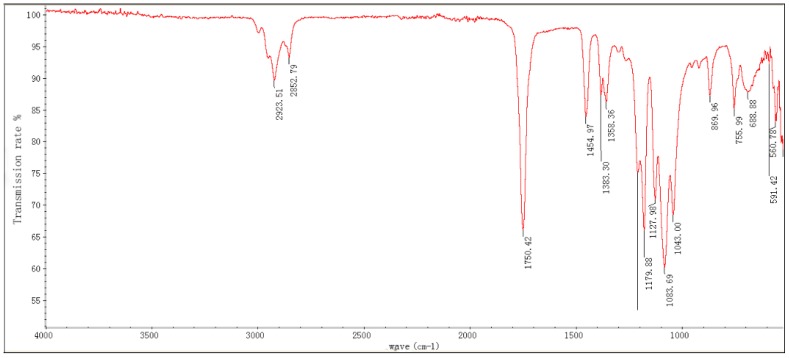
Infrared spectrum of PLA multifilament.

**Figure 5 polymers-11-00065-f005:**
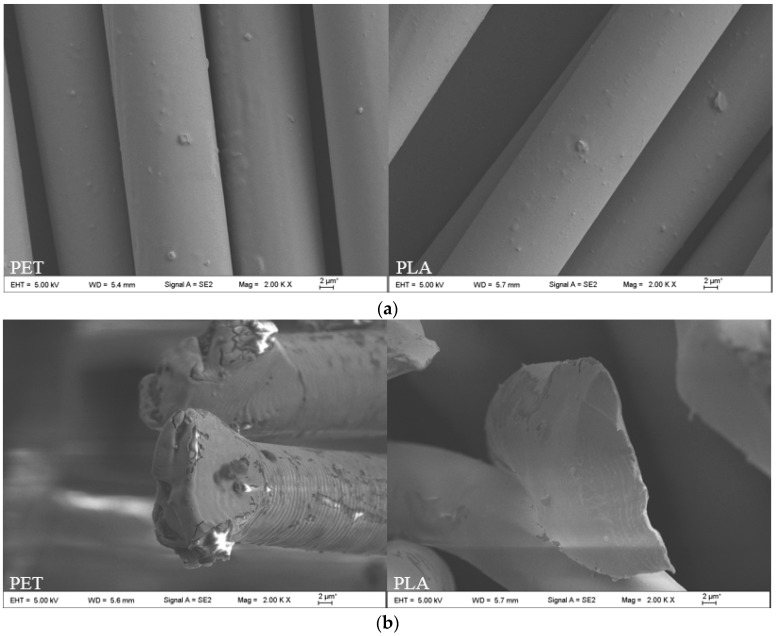
Electron micrograph of PLA multifilament: (**a**) longitudinal section morphology; (**b**) cross-section morphology.

**Figure 6 polymers-11-00065-f006:**
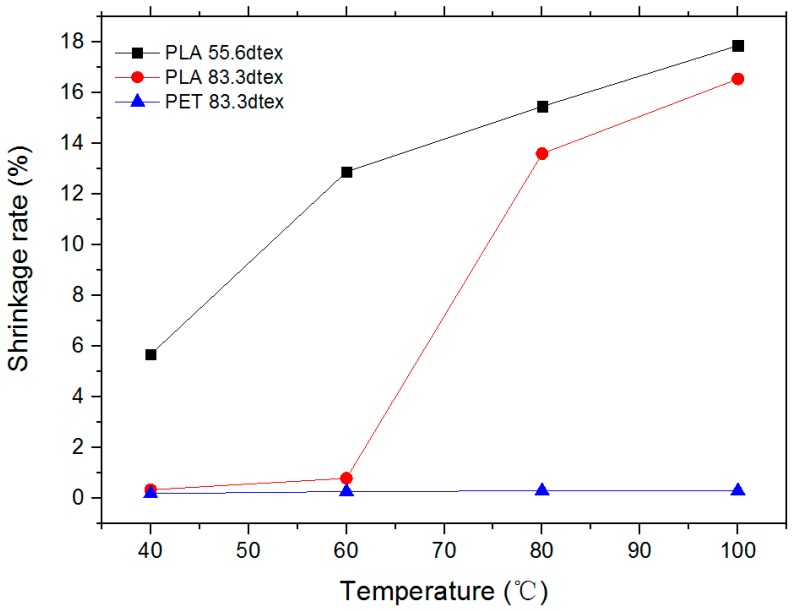
Thermal shrinkage diagram.

**Figure 7 polymers-11-00065-f007:**
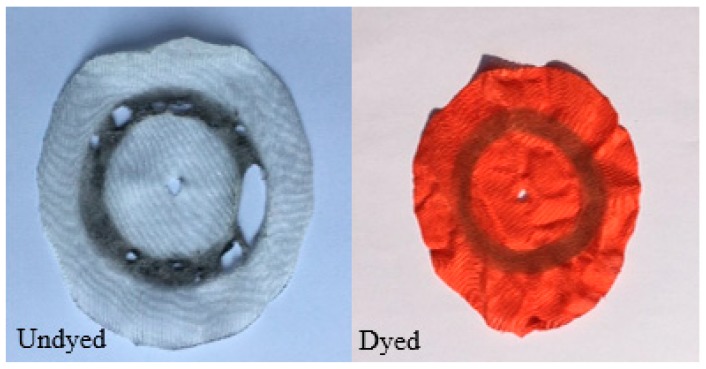
Abrasion resistance of undyed and dyed warp-knitted fabrics.

**Figure 8 polymers-11-00065-f008:**
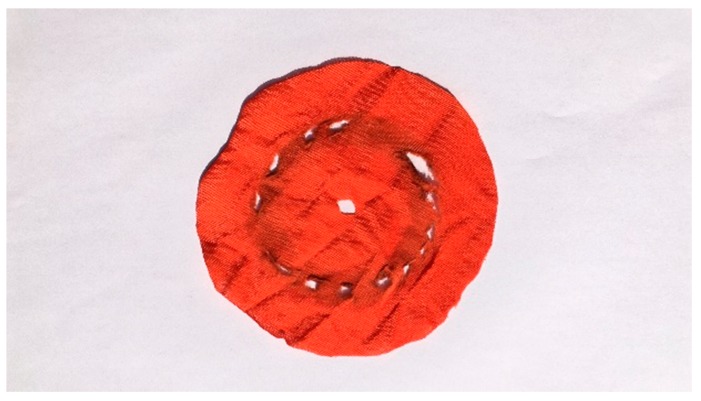
Fabric dyed with no silicone softener.

**Figure 9 polymers-11-00065-f009:**
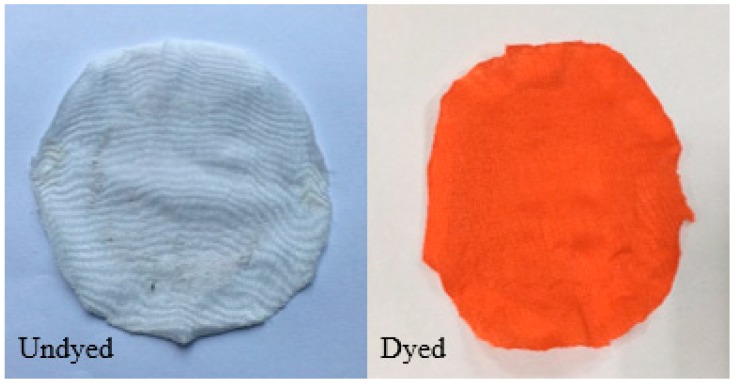
Anti-pilling properties of undyed and dyed warp-knitted fabrics.

**Figure 10 polymers-11-00065-f010:**
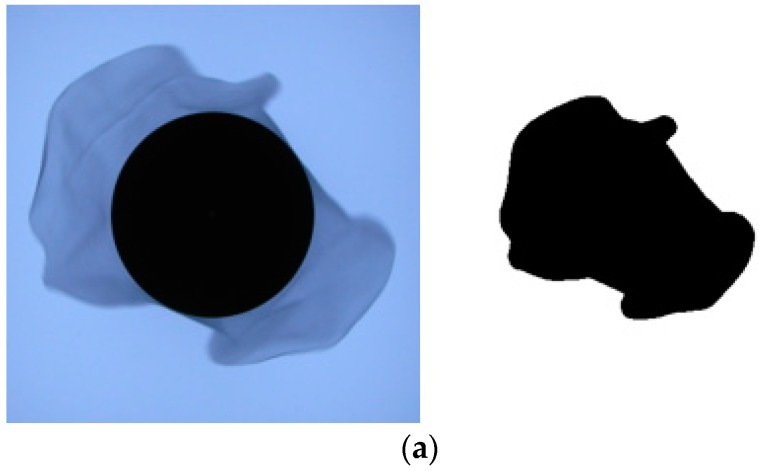
Drape properties of undyed and dyed warp-knitted fabrics: (**a**) Undyed; (**b**) dyed.

**Table 1 polymers-11-00065-t001:** Sample parameter of warp-knitted fabrics.

Material	Filament Diameter/dtex	Structure	Wales/cm	Courses/cm
PLA	83.3	GB1: 1-2/1-0//	11.2	17
		GB2: 1-0/2-3//		

**Table 2 polymers-11-00065-t002:** Mechanical properties of three kinds of multifilament.

Test Index	PLA/55.6 dtex	PLA/83.3 dtex	PET/83.3 dtex
Tensile breaking strength/cN	184.62 ± 2.00	279.61 ± 2.00	290.76 ± 2.00
Tensile breaking strength/cN·dtex^−1^	3.24 ± 0.50	3.35 ± 0.50	3.47 ± 0.50
Elongation at break/%	23.86 ± 1.00	25.01 ± 1.00	34.46 ± 1.00

**Table 3 polymers-11-00065-t003:** Hygroscopic properties of three kinds of multifilament.

Material	Wet Weight/g	Dry Weight/g	Weight Difference/g	Moisture Regain/%
PLA	45.20 ± 1.00	44.96 ± 1.00	0.24	0.53
PET	45.87 ± 1.00	45.68 ± 1.00	0.19	0.42

**Table 4 polymers-11-00065-t004:** Thermal shrinkage rates of three kinds of multifilament.

Temperature/°C	Shrinkage Rate/%		
	PLA/55.6 dtex	PLA/83.3 dtex	PET/83.3 dtex
40	5.68 ± 0.10	0.34 ± 0.10	0.19 ± 0.10
60	12.88 ± 0.10	0.79 ± 0.10	0.27 ± 0.10
80	15.46 ± 0.10	13.60 ± 0.10	0.30 ± 0.10
100	17.86 ± 0.10	16.54 ± 0.10	0.30 ± 0.10

**Table 5 polymers-11-00065-t005:** Mechanical properties of warp-knitted fabrics.

Dyeing State	Tensile Breaking Strength/cN	Elongation at Break/%
Undyed	254.40 ± 5.00	83.19 ± 5.00
Dyed	219.94 ± 5.00	139.42 ± 5.00

**Table 6 polymers-11-00065-t006:** Drape properties of warp-knitted fabrics.

Test Index	Undyed		Dyed	
	Static	Dynamic	Static	Dynamic
Drape coefficient/%	46.06 ± 1.00	47.05 ± 1.00	25.73 ± 1.00	25.84 ± 1.00
Drape/%	23.94 ± 1.00	52.95 ± 1.00	74.27 ± 1.00	74.16 ± 1.00

**Table 7 polymers-11-00065-t007:** Application properties of warp-knitted fabrics.

Dyeing State	Crowe Value/clo	q_max_/cal·cm^2^	Moisture Permeability/g·(m^2^·d)^−^^1^	Air Permeability (mm·s^−1^)
Undyed	0.11 ± 0.05	0.26 ± 0.05	846.0 ± 10.0	2455 ± 10
Dyed	0.16 ± 0.05	0.17 ± 0.05	792.7 ± 10.0	1643 ± 10

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
