# Peer review of "Manufacture and Property of Warp-Knitted Fabrics with Polylactic Acid Multifilament"

_polymers, 2019, doi:10.3390/polym11010065_

Round 1
Reviewer 1 Report
Interesting research work with the findings on warp-knitted fabrics with PLA multifilament of considerable interest. For the benefit of the reader, however, it is better if the authors consider the following mentioned remarks and further improve the manuscript before submitting the final version.
1.Please add a discussion to the two multifilament SEM image, is it affecting subsequent dyeing or other properties?
2.Please explain in detail the trend of the PLA multifilament thermal shrink curve.
3.The physical fabric picture in Figure 2 is not clear
4.Please replace the unit label in Figure 4 with English.
5.Please add further discussion on color fastness in terms of factors affecting the color fastness of the fabric.
Author Response
1. Please add a discussion to the two multifilament SEM image, is it affecting subsequent dyeing or other properties?
Response: The further explanation of the impact on performance and has been marked in blue in the manuscript.
2. Please explain in detail the trend of the PLA multifilament thermal shrink curve.
Response: The detail explain has been add, which is marked in blue in the manuscript.
3. The physical fabric picture in Figure 2 is not clear.
Response: The picture has been replaced.
4. Please replace the unit label in Figure 4 with English.
Response: Figure 4 has been modified in the manuscript.
5. Please add further discussion on color fastness in terms of factors affecting the color fastness of the fabric.
Response: The comprehensive discussion has been added, which is marked in blue in the manuscript.
Special thanks to you for good comments.
Reviewer 2 Report
This paper studied the manufacture and property of warp-knitted fabrics with polylactic acid multifilament. The overall technical and scientific content of this paper is good but improvement is required:
(i) The discussion of mechanical and hygroscopic properties under section "3.1 Properties of PLA multifilament" is not comprehensive. More explanations should be added.
(ii) "wpc" and "cpc" should be written in full in Table 1 should that readers can understand easily.
(iii) Please provide more information about "disperse dye GS/red", e.g. C.I. number, supplier?
(iv) The "Introduction" section is very brief and authors should address the research gap comprehensive in the "Introduction" section.
Author Response
1. The discussion of mechanical and hygroscopic properties under section “3.1 Properties of PLA multifilament” is not comprehensive. More explanations should be added.
Response: The further discussions have been added in the manuscript, which is marked in blue.
2. “wpc” and “cpc” should be written in full in Table 1 should that readers can understand easily.
Response: “wpc” and “cpc” have been written in full in Table 1, Which is marked in blue.
3. Please provide more information about “disperse dye GS/red”, e.g. C.I. number, supplier?
Response: C.I. number and supplier of disperse dyes have been added and marked in blue in the manuscript.
4. The “Introduction” section is very brief and authors should address the research gap comprehensive in the “Introduction” section.
Response: The comprehensive discussion in the “Introduction” section has been added, which is marked in blue in the manuscript.
Special thanks to you for good comments.